# Simultaneous Determination of Ergot Alkaloids in Swine and Dairy Feeds Using Ultra High-Performance Liquid Chromatography-Tandem Mass Spectrometry

**DOI:** 10.3390/toxins13100724

**Published:** 2021-10-13

**Authors:** Saranya Poapolathep, Narumol Klangkaew, Zhaowei Zhang, Mario Giorgi, Antonio Francesco Logrieco, Amnart Poapolathep

**Affiliations:** 1Department of Pharmacology, Faculty of Veterinary Medicine, Kasetsart University, Bangkok 10900, Thailand; fvetsys@ku.ac.th (S.P.); fvetnak@ku.ac.th (N.K.); 2Oil Crops Research Institute of the Chinese Academy of Agricultural Sciences, Wuhan 430062, China; zwzhang@whu.edu.cn; 3Department of Veterinary Science, University of Pisa, 56122 Pisa, Italy; mario.giorgi@unipi.it; 4Institute of Sciences of Food Production, National Research Council, 70126 Bari, Italy; Antonio.logrieco@ispa.cnr.it; 5Center of Excellence on Agricultural Biotechnology (AG-BIO/MHESI), Bangkok 10900, Thailand

**Keywords:** ergot alkaloids, swine feed, dairy feed, UHPLC-MS/MS

## Abstract

Ergot alkaloids (EAs) are mycotoxins mainly produced by the fungus *Claviceps purpurea*. EAs are known to affect the nervous system and to be vasoconstrictors in humans and animals. This work presents recent advances in swine and dairy feeds regarding 11 major EAs, namely ergometrine, ergosine, ergotamine, ergocornine, ergocryptine, ergocristine, ergosinine, ergotaminine, ergocorninine, ergocryptinine, and ergocristinine. A reliable, sensitive, and accurate multiple mycotoxin method, based on extraction with a Mycosep 150 multifunctional column prior to analysis using UHPLC-MS/MS, was validated using samples of swine feed (100) and dairy feed (100) for the 11 targeted EAs. Based on the obtained validation results, this method showed good performance recovery and inter-day and intra-day precision that are in accordance with standard criteria to ensure reliable occurrence data on EA contaminants. More than 49% of the swine feed samples were contaminated with EAs, especially ergocryptine(-ine) (40%) and ergosine (-ine) and ergotamine (-ine) (37%). However, many of the 11 EAs were not detectable in any swine feed samples. In addition, there were contaminated (positive) dairy feed samples, especially for ergocryptine (-ine) (50%), ergosine (-ine) (48%), ergotamine (-ine), and ergocristine (-ine) (49%). The mycotoxin levels in the feed samples in this study almost complied with the European Union regulations.

## 1. Introduction

Mycotoxins are hazardous chemicals produced by *Aspergillus*, *Fusarium Penicillium,* and *Claviceps* genus. Mycotoxins can contaminate foods and feeds and agricultural products [1]. To date, there are more than four hundred mycotoxins with different toxicity, which have been identified in cereals, fruits, vegetables, and other agricultural commodities, resulting in potential adverse effects on human and animal health, and economic losses [2,3,4]. Moreover, mycotoxins are persistent in food and feeds and not completely eliminated during processing operations [3]. Recently, mycotoxins were a major category in border rejection in the European Union (EU) according to the annual report of the Rapid Alert System for Food and Feed (RASFF) [3]. The Food and Agriculture Organization (FAO) suggested one-fourth of global food crops is contaminated by mycotoxins [5]. Because of their pathogenicity and lethality, worldwide authorities including the World Health Organization (WHO) have called to monitor mycotoxins in foodstuff and feeds and set up strict maximum levels and legislation, in order to provide an early warning about mycotoxin contamination and reduce the national and international losses. In addition, the impact of climate change on *Calviceps* spp. infection of crops could result in a potential to increase the higher food safety risks for humans and animals due to mycotoxin contamination in the end products [6].

Ergot alkaloids (EAs) are toxic secondary metabolites produced by fungi of the *Claviceps* genus, mainly by the parasitic fungus *Claviceps purpurea,* which parasitize the seed heads of living plants at the time of flowering [7]. EAs are known to cause adverse health effects in humans and animals and have been found in cereals, cereal products, barley, oats, and both rye- and wheat-containing foods [8,9,10,11]. Outbreaks of ergotism in livestock do still occur, and EAs can induce abortion by its toxicity [12]. Pigs and cattle have shown symptoms after being infected with EAs, causing financial problems to both breeders and the meat industry [12,13]. Animals, including pigs exposed to EAs from grains, can cause liver and intestinal alterations [14]. In Directive 2002/32/EC on undesirable substances in animal feed and its amendment, the maximum content of rye ergot (*Claviceps purpurea*) in feed containing unground cereals has been established at 1000 mg/kg. EAs have been reported in cereals in European countries, Canada, the United States, and China [15,16,17]. There have also been some reports on the presence of EAs in feed from other countries, with 86–100% of EAs detected in feed samples from Germany [18] and 83% of compound feeds containing EAs with an average concentration of 89 μg/kg and a maximum concentration of 1231 μg/kg in the Netherlands [19]. The main ergot alkaloids produced by *Claviceps* species are ergometrine, ergotamine, ergosine, ergocristine, ergokryptine, and ergocornine, and the group of agroclavines [20]. Ergotamine and ergosine are heat stable whereas ergocristine, ergokryptine, ergocornine, and ergometrine are decreased by heating [21]. The conversion of ergopeptines to ergopeptinines was accelerated either by acidic or alkaline solutions. However, ergopeptinines can also be transformed to ergopeptines in organic solvents [7,22].

Studies have developed reliable analytical methods of EAs in agricultural commodities [12,15,17,19,22,23,24,25,26], mainly using HPLC-MS/MS. However, the challenge remains in the UHPLC-MS/MS method of optimizing the sample preparation procedure. However, signal suppression and enhancement usually occur due to the interferences in the matrix (matrix effect), leading to unreliable results [25]. To compensate for the matrix effect, some methods developed for the analysis of EAs in agricultural commodities have used a MycoSep^®^ multifunctional column [2].

To the best of the authors’ knowledge, to date, there have been a few reports on contaminations of EAs in any kinds of foodstuffs and feeds in Thailand. The current study investigated the occurrence of 11 EAs in swine and dairy feeds using a validated UHPLC-MS/MS with a multifunctional SPE column procedure. We used an SPE column for sample cleanup. Under optimization, the limit of detection, limit of quantification and linearity were studied. Accuracy and precision were evaluated as well. This work provides a promising manner to monitor EAs in feed samples.

## 2. Results and Discussion

### 2.1. Method Validation

The results of the limit of detection, limit of quantification, and linearity are reported in Table 1. From this study, the method produced good linearity.

Over the relevant working range, the calibration curve showed good linearity with the *r*^2^ value higher than 0.995. The LOD value was 0.25 ng/g, and the LOQ was 0.5 ng/g (Table 1). The recovery and precision values were 70–120%, and the % relative standard deviation (RSD) values were less than 20% [27] for all 11 ergot alkaloids, as summarized in Table 2 and Table 3 for the swine and dairy feeds, respectively. For identification requirements, the relative ion ratio from sample extracts was lower than 30% for all 11 ergot alkaloids [27].

### 2.2. Matrix Effect Study

The study used % signal suppression/enhancement (SSE) to evaluate the matrix effects in the two types of feed matrices. If the suppression or enhancement were marginal, the %SSE would be very close to 100%; if there was strong suppression or enhancement, the %SSE would deviate from 100%. In the swine feed samples, the %SSE (94.5–106.7%) was within the acceptable range (80–120%SSE), except for ergometrine, which exhibited strong signal suppression with its %SSE (75.1%) below the acceptable range. In the dairy feed samples, the %SSE for signal suppression for the 11 ergot alkaloids was within the acceptable range 83.8–98.1%, except for ergotamine and ergometrine, which exhibited strong signal suppression (%SSE 79.6% and 44.5%, respectively). The %SSE values of the two types of feed matrices are summarized in Figure 1. For all the results of the matrix effect, the quantification of the 11 ergot alkaloids using matrix-matched calibration is necessary. The extract ion chromatograms (XIC) of spiked 11 EAs in swine and dairy feed samples were illustrated in Figure 2 and Figure 3, respectively.

### 2.3. Occurrence of EAs in Swine and Dairy Feeds

The method derived from this study was applied to explore the 11 ergot alkaloids in 200 feed samples consisting of swine (*n* = 100) and dairy feeds (*n* = 100). In the swine feed samples, more than 49% were contaminated with ergot alkaloids, especially ergocryptine (-ine) (40%), ergosine (-ine), and ergotamine (-ine) (37%). However, more than 50% of total samples were not detectable in 11 ergot alkaloids in the swine feed sample. The dairy feed samples had the same prevalent contaminants as the swine feed samples but with higher positive samples, especially for ergocryptine (-ine) (50%), ergosine (-ine) (48%), ergotamine (-ine), and ergocristine (-ine) (49%), as shown in Table 4 and Table 5. The mycotoxin levels in all feed samples almost complied with the EU regulation (≤1000 mg/kg of 11 ergot alkaloids) [28]. There are several reports on the presence of EAs in the feed from different countries, with 86–100% of listed EAs detected in feed samples from Germany [1] and 83% of compound feeds containing EAs with an average concentration of 89 μg/kg and a maximum concentration of 1231 μg/kg in the Netherlands [19]. The major detected EAs were ergosine, ergotamine, ergocristine, and ergocryptine. Interestingly, Malysheva et al. [13] reported the occurrence of EAs over three years in 1065 cereal samples originating from 13 European countries, with 52% of rye, 27% of wheat, and 44% of total samples containing EAs (ergosine, ergocristine, and ergocryptine) ranging from less than 1 to 12,340 μg/kg. In Spain, the concentrations for individual ergot alkaloids ranged between 5.9 μg/kg for ergosinine to 145.3 μg/kg for ergometrine, while the total ergot alkaloid content ranged from 5.9 to 158.7 μg/kg in swine samples. About 12.7% revealed contamination by at least one ergot alkaloid, and among contaminated swine samples, 65% were contaminated by more than one [22].

The ergot contaminations and patterns were differences due to the geographical region and environmental conditions [10].

## 3. Conclusions

EAs are hazardous mycotoxins in food and feed samples. Our results showed that the LC-ESI-MS/MS technique was an excellent tool for untargeted determination of 11 EAs in swine and dairy feed samples. The validated LC-MS/MS method using a multifunctional column was successfully performed according to the SANTE/11813/2017 standard. LODs and LOQs were recorded as 0.25 and 0.5 ng/g for EAs. Recoveries were 90.6–120%. When this technique was applied to real feed samples, it showed that 11 EAs were quantifiable in animal feeds. The mycotoxin levels in the swine and dairy samples almost complied with the EU regulations. The presence of ergot sclerotia is regulated to a maximum of 500 mg/kg in unprocessed cereal for humans [29] and 1000 mg/kg in feed materials and compound feed containing unground cereals [30]. However, further studies with a larger sample size are needed to confirm these as acceptable levels. The knowledge of toxigenic *Claviceps* species for better understanding of the production of EAs and to progress appropriate solutions for disease management should be investigated.

## 4. Materials and Methods

### 4.1. Reagents and Materials

The LC-MS/MS grade reagents, consisting of ammonium carbonate and acetonitrile (MeCN), were purchased from Fluka (St. Louis, MO, USA). The Mycosep 150 multifunctional column for extraction clean-up was purchased from Romer Labs (Tulln, Austria). Deionized water was produced using a Milli-Q system (Millipore; Bedford, MA, USA).

### 4.2. Analytical Standards

The analytical standards of the ergot alkaloids (ergometrine, ergosine, ergotamine, ergocornine, ergocryptine, ergocristine, ergosinine, ergotaminine, ergocorninine, ergocryptinine, and ergocristinine) were purchased from Chiron (Trondheim, Norway).

### 4.3. Preparation of Standards Solution

The analytical standard ergot alkaloid stock solutions were prepared in acetonitrile to provide a working standard solution of 100 µg/mL concentration for ergometrine, ergosine, ergotamine ergocryptine, ergocristine, and ergocornine and 25 µg/mL for ergosinine, ergotaminine, ergocryptinine, ergocristinine, and ergocorninine. For method validation of the spiking experiments, working standard solutions were freshly prepared at 1.0 µg/mL and were stored in amber vials at −20 °C for one week.

### 4.4. Sample Collection

A total of 200 feed samples consisting of swine feed (*n* = 100) and dairy feed (*n* = 100) were randomly collected from animal farms in different regions of Thailand. All samples were ground in a rotor mill ZM200 (Retsh GmbH, Hann, Germany) into a fine powder (0.50 mm) and stored at −20 °C before analysis.

### 4.5. Sample Preparation

The sample preparation protocol applied was developed based on Krska et al. [10]. Briefly, 5 g of homogenized feed sample was weighed into a 50 mL polypropylene (PP) centrifugation tube, followed by the addition of 25 mL of acetonitrile–ammonium carbonate buffer (3.03 mM), 84:16 (*v/v*). The tube was closed and shaken using a laboratory shaker (IKA Labortechnik; Staufen, Germany) for 30 min at 240 rpm. The extract was passed through Whatman No. 4 filter paper, and 4 mL of the extract was transferred to the Mycosep 150 multifunctional column (Romers lab, Tulln, Austria). Then, 1 mL of the purified extract was evaporated to dryness at 40 °C. The residue was reconstituted in 500 μL 50% mobile phase, and the mixture was passed through a 0.22 μm nylon filter before being used in the LC-MS/MS analysis.

### 4.6. UHPLC-MS/MS Analysis

The 11 target ergot alkaloids were analyzed using the UHPLC-MS/MS method. Chromatographic separation was developed according to Krska et al. [10]. The analysis used a Shimadzu LC-MS 8060 system (Shimadzu, Tokyo, Japan) that was equipped with a Gemini analytical column (150 × 2.0 mm i.d., 5.0 μm particle size; Phenomenex; Torrance, CA, USA) maintained at 30 °C. The mobile phase for analyses used 3.03 mM ammonium carbonate in deionized water (A) and MeCN (B) in ESI (+). The gradient elution was identical initially. The proportion of B was immediately increased from 5% to 17% within 1 min and further linearly increased to 47%, 54%, and 80% after 2, 10, and 15 min, respectively. Subsequently, the proportion of B was decreased to the initial conditions (5%) over 1 min, followed by a hold-time of 5 min, resulting in a total run-time of 21 min. The flow rate was stable at 0.5 mL/min throughout the run; 10 μL of sample extract was injected into the LC-MS/MS system.

The Shimadzu LC-MS 8060 system (Shimadzu, Japan) was equipped with an electrospray (ESI) ion source operated in positive mode. The ion source parameters were a nebulizing gas flow of 3 L/min, a heating gas flow of 10 l/min with an interface temperature: 300 °C, a CDL temperature of 250 °C, a heating block temperature of 400 °C, and a drying gas flow of 10 L/min. The dwell time (ms), Q1 Pre Bias (V), CE (V), and Q3 Pre Bias (V) were optimized during infusion of individual analytes (100 ng/mL) using automatic infusion. The MRM transitions of 11 ergot alkaloid-dependent parameters are summarized in Table 6.

### 4.7. Method Validation Procedure

The method performance characteristic parameters was determined to assess the efficiency of analytical method from this study by evaluating the linearity, accuracy, precision, LOD, and LOQ for EA contamination in swine and dairy feed samples. The analytes were quantified using a matrix-matched calibration standard with a pre spiking calibration curve for the 11 EAs for levels in the range 0.5–100.0 ng/g. The accuracy and precision (%RSD) were determined within the day by analyzing five replicates at three levels. The inter-day precision was determined at the same level as the within-day precision on three different days (*n* = 15). LODs and LOQs were calculated by analyzing the spiked samples at low level concentrations. LODs were determined as the lowest concentration of the analyte for which a signal-to-noise (S/N) ratio was 3:1, whereas S/N ratio was 10:1 for LOQs.

### 4.8. Matrix Effects Study

The matrix effects of the method were evaluated within two types of feed matrices: swine and dairy feed. Matrix-matched calibration curves were prepared at seven levels in the range 0.5–100.0 ng/g (*n* = 3 per each concentration). The matrix effects expressing the matrix-induced SSE% were defined as percentage ratios of the matrix-matched calibration slope to the solvent calibration slope. Therefore, the matrix-matched calibration curves were used for quantitative analysis.

## Figures and Tables

**Figure 1 toxins-13-00724-f001:**
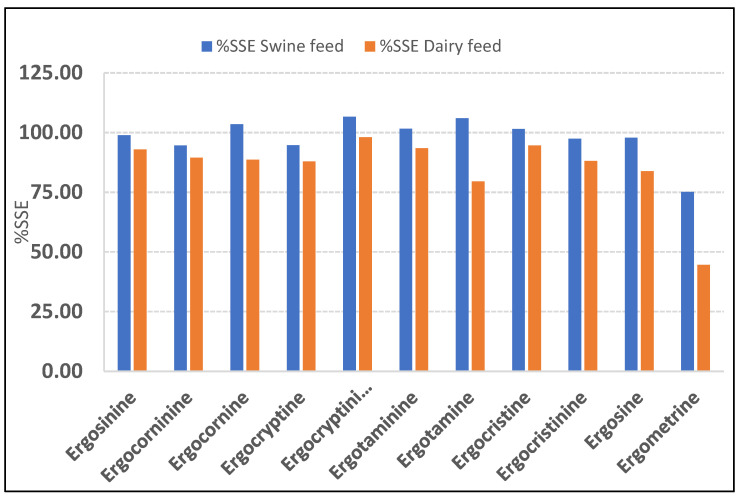
Signal suppression/enhancement (%SSE) for 11 ergot alkaloids in matrix-matched calibration.

**Figure 2 toxins-13-00724-f002:**
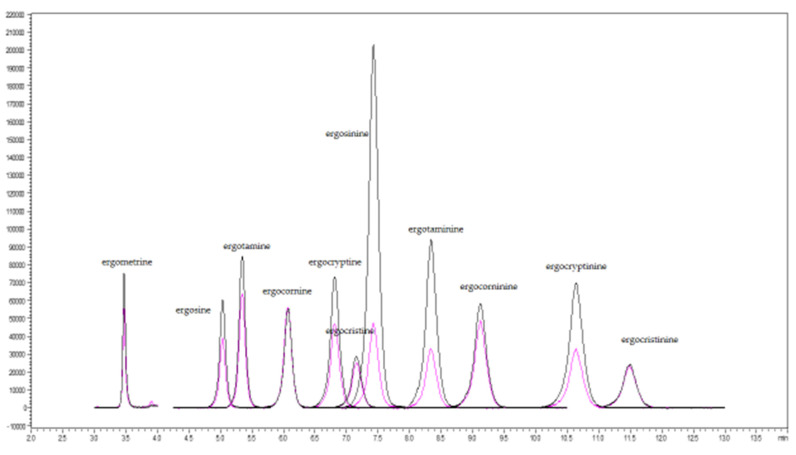
Extracted ion chromatogram (XIC) of spiked 11 ergot alkaloids at 20 ng/g in swine feed samples.

**Figure 3 toxins-13-00724-f003:**
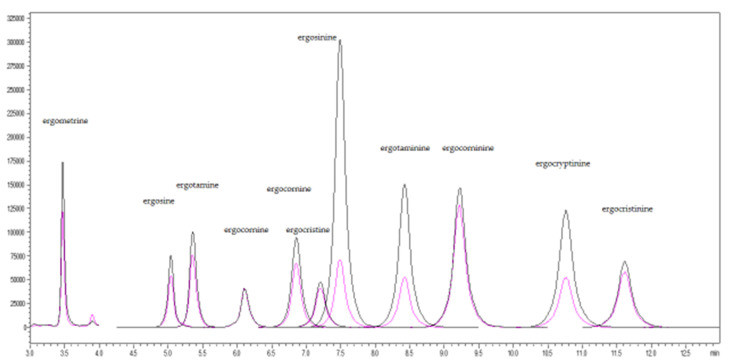
Extracted ion chromatogram (XIC) of spiked 11 ergot alkaloids at 20 ng/g in dairy feed samples.

**Table 1 toxins-13-00724-t001:** Performance characteristic of the analytical method: linearity ranges, limit of detection (LOD), and limit of quantification (LOQ) of the optimized LC-MS/MS method for simultaneous determination of 11 ergot alkaloids.

Ergot Alkaloid	LOD (ng/g)	LOQ (ng/g)	Calibration Range (ng/g)
Ergometrine	0.25	0.5	0.5–100
Ergosine	0.25	0.5	0.5–100
Ergocornine	0.25	0.5	0.5–100
Ergocryptine	0.25	0.5	0.5–100
Ergocristine	0.25	0.5	0.5–100
Ergotamine	0.25	0.5	0.5–100
Ergosinine	0.25	0.5	0.5–100
Ergocorninine	0.25	0.5	0.5–100
Ergocryptinine	0.25	0.5	0.5–100
Ergocristinine	0.25	0.5	0.5–100
Ergotaminine	0.25	0.5	0.5–100

**Table 2 toxins-13-00724-t002:** Accuracy and precision study for 11 ergot alkaloids determination in optimal LC-MS/MS conditions for swine feed samples.

Ergot Alkaloids	Spike Level, (ng/g)	Swine Feed		
		%Recovery, (%)	Intra-Day Precision, (%RSD)	Inter-Day Precision, (%RSD)
Ergometrine	0.5	113.1	2.60	8.8
	10.0	94.2	1.33	7.1
	100.0	96.2	2.95	4.7
Ergosine	0.5	111.7	5.60	8.14
	10.0	115.8	2.87	11.59
	100.0	109.2	2.88	13.38
Ergocornine	0.5	105.3	3.55	4.80
	10.0	115.1	9.39	15.69
	100.0	116.6	3.60	8.45
Ergocryptine	0.5	118.9	4.21	7.22
	10.0	109.0	11.44	11.62
	100.0	114.4	2.73	6.87
Ergocristine	0.5	107.1	8.77	9.34
	10.0	119.6	14.07	16.63
	100.0	120.0	7.45	8.95
Ergotamine	0.5	116.9	2.70	9.76
	10.0	117.3	8.68	15.67
	100.0	117.1	5.52	9.90
Ergosinine	0.5	99.1	2.25	6.51
	10.0	98.1	2.29	5.99
	100.0	97.0	1.96	5.09
Ergocorninine	0.5	101.9	4.08	8.48
	10.0	100.7	5.74	5.52
	100.0	100.0	3.07	5.84
Ergocryptinine	0.5	110.9	2.84	5.86
	10.0	106.7	4.52	10.13
	100.0	100.2	3.52	6.11
Ergocristinine	0.5	111.3	3.96	8.71
	10.0	101.6	4.44	5.33
	100.0	98.8	1.77	6.04
Ergotaminine	0.5	100.5	3.30	7.19
	10.0	97.6	2.88	8.93
	100.0	97.0	1.64	5.69

%RSD = percentage relative standard deviation.

**Table 3 toxins-13-00724-t003:** Accuracy and precision study for 11 ergot alkaloids determination in optimal LC-MS/MS conditions for dairy feed samples.

Ergot Alkaloid	Spike Level, (ng/g)	Dairy Feed		
		%Recovery, (%)	Intra-Day Precision, (%RSD)	Inter-Day Precision, (%RSD)
Ergometrine	0.5	92.1	2.11	8.2
	10.0	101.1	1.40	7.2
	100.0	97.4	4.77	6.8
Ergosine	0.5	102.0	7.10	6.33
	10.0	103.0	2.22	4.55
	100.0	101.0	1.95	3.08
Ergocornine	0.5	99.6	2.98	7.78
	10.0	99.4	4.18	5.49
	100.0	92.7	4.33	10.79
Ergocryptine	0.5	101.9	2.67	6.14
	10.0	98.2	2.97	5.01
	100.0	91.5	4.10	11.83
Ergocristine	0.5	102.1	1.09	4.09
	10.0	98.1	5.51	7.02
	100.0	90.6	3.24	13.29
Ergotamine	0.5	102.4	4.13	7.85
	10.0	103.0	5.19	5.54
	100.0	101.3	1.91	2.18
Ergosinine	0.5	100.3	3.49	3.98
	10.0	100.0	1.92	3.77
	100.0	101.4	1.70	4.47
Ergocorninine	0.5	95.7	3.21	9.92
	10.0	97.7	0.97	2.32
	100.0	98.1	2.02	3.46
Ergocryptinine	0.5	97.1	2.57	8.42
	10.0	99.1	1.13	2.85
	100.0	100.4	1.82	3.34
Ergocristinine	0.5	102.7	4.51	6.73
	10.0	95.9	3.84	2.93
	100.0	98.9	1.72	3.55
Ergotaminine	0.5	101.3	1.39	4.39
	10.0	100.0	3.13	5.79
	100.0	100.3	2.05	6.73

%RSD = percentage relative standard deviation.

**Table 4 toxins-13-00724-t004:** Occurrence of 11 ergot alkaloids in swine feed samples.

Ergot Alkaloid	Swine Feed (*n* = 100)		
	Number of Positive Samples	Range (ng/g)	Mean (ng/g)
Ergosinine	37	0.53–9.72	2.06
Ergosine	30	0.40–4.99	1.57
Ergocorninine	26	0.46–25.25	4.99
Ergocornine	23	0.29–4.82	1.83
Ergocryptinine	40	0.25–100.55	7.64
Ergocryptine	17	0.63–17.22	4.41
Ergotaminine	37	0.27–13.46	2.96
Ergotamine	33	0.31–18.5	3.14
Ergocristinine	28	0.67–77.6	16.15
Ergocristine	28	0.57–48.00	9.16
Ergometrine	20	0.52–10.87	3.06

**Table 5 toxins-13-00724-t005:** Occurrence of 11 ergot alkaloids in dairy feed samples.

Ergot Alkaloid	Dairy Feed (*n* = 100)		
	Number of Positive Samples	Range (ng/g)	Mean (ng/g)
Ergosinine	48	0.52–16.61	2.69
Ergosine	36	0.45–12.17	2.00
Ergocorninine	46	0.38–43.60	6.25
Ergocornine	35	0.31–11.47	2.44
Ergocryptinine	50	0.44–31.57	8.25
Ergocryptine	33	0.58–13.19	3.52
Ergotaminine	49	0.46–52.11	6.04
Ergotamine	48	0.34–43.02	5.34
Ergocristinine	49	0.62–210.53	26.75
Ergocristine	47	0.26–98.19	13.05
Ergometrine	36	0.26–31.67	2.89

**Table 6 toxins-13-00724-t006:** MS/MS parameters for determination of 11 ergot alkaloids.

Analyte	*m/z*	Dwell Time (ms)	Q1 Pre Bias (V)	CE (V)	Q3 Pre Bias (V)	Retention Time (Min)
Ergocorninine	562.40 > 223.30	70.0	−22.0	−34.0	−15.0	
	562.40 > 277.30	70.0	−26.0	−29.0	−19.0	9.4
Ergocornine	562.35 > 223.30	60.0	−22.0	−37.0	−24.0	
	562.35 > 208.20	60.0	−22.0	−45.0	−23.0	6.15
Ergocryptine	576.40 > 223.30	60.0	−22.0	−35.0	−25.0	
	576.40 > 208.30	60.0	−22.0	−49.0	−22.0	6.93
Ergocryptinine	576.35 > 223.30	80.0	−22.0	−37.0	−16.0	
	576.35 > 208.20	80.0	−22.0	−52.0	−23.0	10.99
Ergotaminine	582.30 > 223.30	70.0	−22.0	−34.0	−16.0	
	582.30 > 277.25	70.0	−22.0	−26.0	−20.0	8.57
Ergotamine	582.30 > 223.30	60.0	−22.0	−33.0	−16.0	
	582.30 > 208.20	60.0	−22.0	−44.0	−23.0	5.38
Ergocristine	610.40 > 223.30	60.0	−24.0	−36.0	−25.0	
	610.40 > 208.25	60.0	−24.0	−47.0	−22.0	7.29
Ergocristinine	610.40 > 223.30	60.0	−28.0	−36.0	−16.0	
	610.40 > 325.30	60.0	−24.0	−28.0	−22.0	11.85
Ergosine	548.45 > 223.10	60.0	−40.0	−33.0	−16.0	
	548.45 > 208.25	60.0	−40.0	−40.0	−14.0	5.05
Ergosinine	548.35 > 223.30	80.0	−20.0	−32.6	−16.0	
	548.35 > 263.10	80.0	−20.0	−27.8	−19.0	7.6
Ergometrine	326.30 > 223.30	60.0	−24.0	−25.0	−25.0	
	326.30 > 208.20	60.0	−24.0	−30.0	−22.0	3.46

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
