# Peer review of "Simultaneous Determination of Ergot Alkaloids in Swine and Dairy Feeds Using Ultra High-Performance Liquid Chromatography-Tandem Mass Spectrometry"

_toxins, 2021, doi:10.3390/toxins13100724_

Round 1
Reviewer 1 Report
This work present the ocurrence study of 11 EAs in swine and dairy feeds using a UHPLC-MS/MS method for their determination. This work have been clearly and properly presented. I recommend to accept this work due to the lack of studies carried out for EAs in feed samples and because of the recommendations by the EFSA in which more studies about this topic are requested. However, several points should be improved.
Some considerations and questions for the authors:
- I encourage the authors to empathize the reason of the origin of the samples as well as the kind of samples selected due to lack of studies involving them. Why is it important to study these samples in Thailand?
- I strongly recommend the authors to consider other title for this manuscript since the interest of this work is mainly based on the ocurrence study and not in the analytical technique used (previously reported in other work).
- A chromatogram showing the separation and identification of each EAs in a spiked feed sample must be provided, as well as the retention times for all of them in table 6.
- Why did you not include the epimer of ergometrine (ergometrinine) in your study?
- I do not understand the sentence "matrix-matched calibration standard with a pre spiking calibration curve". Do you mean procedural calibration?. Please, explain this point. How did you carry out the calibration curve? Did you do calibration curves for each matrix?
- how many samples did you treat to calculate %SSE (n=?)?
- the same for the precision study. Indicate the number of experimental and instrumental replicates.
- the reference 22 should be included and commented in section 2.3 for being the most recent work including this sort of samples and EAs.
other corrections:
- Genera-->genus? in line 24
- define WHO in line 33
- Correct ºC in line 167
Author Response
Dear Reviewers;
We are pleased and grateful to the reviewers recognizing that he topic of the paper matches with the scope of the Toxins. The authors would like to thank you for your time and feedback on our manuscript and provide the following responses addressing your comments.
Reviewer 1#
Some considerations and questions for the authors:
- I encourage the authors to empathize the reason of the origin of the samples as well as the kind of samples selected due to lack of studies involving them. Why is it important to study these samples in Thailand?
In agreement with the reviewer’s comment, it has been emphasized why it is important to study these samples in Thailand in the revised manuscript.
“To the best authors’ knowledge, to date, there have been a few reports on contaminations of EAs in any kinds of foodstuffs and feeds in Thailand.” l 70-71
- I strongly recommend the authors to consider other title for this manuscript since the interest of this work is mainly based on the occurrence study and not in the analytical technique used (previously reported in other work).
The title was changed to “Occurrence of major ergot alkaloids in dairy and swine feeds using ultra high performance liquid chromatography- tandem mass spectrometry” L 1-4
- A chromatogram showing the separation and identification of each EAs in a spiked feed sample must be provided, as well as the retention times for all of them in table 6.
The extracted ion chromatograms (XIC) figure was added in the revised manuscript. (Figure 2 & 3) Pages 6 & 7
In addition, the retention times of 11 EAs were added in the revised manuscript. (Ersosinine = 7.6 min, Ergocorninine = 9.4 min, Ergocornine = 6.15 min, Ergocryptine = 6.93 min, Ergocryptinine = 10.99 min
Ergotaminine = 8.57 min, Ergotamine = 5.38 min, Ergocristine = 7.29 min, Ergocristinine = 11.85 min, Ergosine = 5.05 min), Ergometrine = 3.46 min (See in Table 6).
- Why did you not include the epimer of ergometrine (ergometrinine) in your study?
In line with the reviewer’s comment, ergometrinine should be analyzed in dairy and swine feeds, but it has not been explored in this study. However, we will keep this suggestion for the next study.
- I do not understand the sentence "matrix-matched calibration standard with a pre spiking calibration curve". Do you mean procedural calibration? Please, explain this point. How did you carry out the calibration curve? Did you do calibration curves for each matrix?
We would like to clarify that the standard mixture of 11 EAs was added to each matrix (dairy and swine feeds) to create matrix-matched calibration standard, in order to quantify 11 EAs in the feed sample.
- How many samples did you treat to calculate %SSE (n=?)?
The authors used two types of feeds, and selected one sample to calculate %SSE (7 point of standard solution calibration curves compared with matrix matched calibration curves). (n= 3 per each concentration) The information was added in the revised manuscript. (Section 5.8)
- The same for the precision study. Indicate the number of experimental and instrumental replicates.
For the precision study, the intra-day precision (%RSD) were determined within the day by analyzing five replicates at three levels. The inter-day precision was determined at the same level as the within-day precision on three different days (n= 15). The information was added in the revised manuscript. (Section 5.7)
- The reference 22 should be included and commented in section 2.3 for being the most recent work including this sort of samples and EAs.
In agreement with the reviewer’s comment, the information from ref. 22 was added in the revised manuscript.
“In Spain, the concentrations for individual ergot alkaloids ranged between 5.9 g/kg for ergosinine to 145.3 mg/kg for ergometrine, while the total ergot alkaloid content ranged from 5.9 to 158.7 mg/kg in swine samples. About 12.7% revealed contamination by at least one ergot alkaloid, and among contaminated in swine samples, 65% were contaminated by more than one [22].” L 139-143
Other corrections:
- Genera-->genus? in line 24
It was rewritten. (L 24)
- define WHO in line 33
The full name of WHO was spelt out. (L 33)
- Correct ºC in line 167
It was corrected.
With best regards,
Authors

Reviewer 2 Report
Dear authors
Correction and elaboration
par. 121 and 141: 'almost complied with the EU regulations'. This statement is imprecise and needs elaborating. It is worth to mention exactly how much the norm was exceeded. Are the tested feeds admitted to trading in EU? Maybe (but not necessary) can the research and the described method of testing influence swine and diary feeds industry production? Are the norms exceeded to high and worth worrying about? Just a few more statements in the conclusions about possible significance of the research would be great.
Minor corrections:
par. 23: Check the use of 'metabolized'. Maybe there should be 'produced' (?)
par. 24: Usage of 'ready to' not clear. It is hard to understand
par. 37: 'in a substantial increase' - increase of what?
par. 45: 'these compounds have also been used as abortifacients' - by whom? Rather not by breeders on livestock as implied by the phrase itself :)
Very interesting paper!
Best regards
Author Response
Dear Reviewers;
We are pleased and grateful to the reviewers recognizing that he topic of the paper matches with the scope of the Toxins. The authors would like to thank you for your time and feedback on our manuscript and provide the following responses addressing your comments.
Reviewer 2#
Correction and elaboration
- par. 121 and 141: 'almost complied with the EU regulations'. This statement is imprecise and needs elaborating. It is worth to mention exactly how much the norm was exceeded. Are the tested feeds admitted to trading in EU? Maybe (but not necessary) can the research and the described method of testing influence swine and diary feeds industry production? Are the norms exceeded to high and worth worrying about? Just a few more statements in the conclusions about possible significance of the research would be great.
In agreement with the reviewer’s comment, the paragraph was rewritten for EU regulations.
“The presence of ergot sclerotia is regulated to a maximum of 500 mg/kg in unprocessed cereal for humans [12] and 1000 mg/kg in feed materials and compound feed containing unground cereals [13]” L 156-157
- European Commission. Commission Regulation (EC) No 1881/2006 of 19 December 2006 setting maximum levels for certain contaminants in foodstuffs. Off. J. Eur. Union 2006, L364, 5–24. L 299-300
- European Commission. Directive 2002/32/EC of the European Parliament and of the Council of 7 May 2002 on undesirable substances in animal feed. Off. J. Eur. Communities 2002, L140, 10–21. L 301-303
Minor corrections:
- par. 23: Check the use of 'metabolized'. Maybe there should be 'produced' (?)
The word was amended in the revised manuscript accordingly. (L 23)
- par. 24: Usage of 'ready to' not clear. It is hard to understand.
The statement was rewritten in the revised manuscript. (L 24)
- par. 37: 'in a substantial increase' - increase of what?
- The statement was rewritten in the revised manuscript. (L 37)
- par. 45: 'these compounds have also been used as abortifacients' - by whom? Rather not by breeders on livestock as implied by the phrase itself :)
The statement was rewritten in the revised manuscript. (L 45)
Very interesting paper!
With many- many thanks for kindly consideration.
Best regards,
Authors

Round 2
